# WidgetEval: Benchmarking Foundation Models on Dynamic Widget Generation for Apps

## Abstract

In this paper, we study the problem of generating application widgets on the fly based on context and user specifications. A widget in this setting encapsulates application APIs while providing a seamless user experience that integrates with the host application. We focus on domains such as Microsoft Excel, Microsoft Word, and synthetic environments such as calendar, file system, and messenger applications. Widget generation is a challenging problem that requires foundation models to understand APIs, write code, reason about application context, and design user interfaces. To understand foundation models' ability to generate on-demand widgets, we first establish a dataset of widget tasks, test scenarios, and functionality verification mechanisms for various applications. We then devise an evaluation strategy which simulates user interaction with the generated widget and checks for runtime and functionality errors. We compare popular models with various prompting and generation strategies. Overall, we found closed source models such as GPT-4o, GPT-4.1 and o4-mini outperform models such as LLaMa-3.1-70B and Phi4, but even with the best configuration, their success rates on Excel and Word widget tasks remain around 50%.

## 1 Introduction

Foundation models like GPT-4o (OpenAI, 2024), GPT-4.1 (OpenAI, 2025a), Claude (Anthropic, 2025), and Gemini (Google, 2025) has shown strong capabilities in code generation, opening new opportunities for generating user interfaces (GenUI) (Lu et al., 2024b; Chen et al., 2025b). The scope of GenUI applications is broad, ranging from writing full applications from scratch (Lu et al., 2025), to transforming hand-drawn sketches to UI prototypes (Si et al., 2024), to generating widgets incrementally (Vaithilingam et al., 2024) and designing custom interfaces for presenting LLM's outputs and followup questions (Chen et al., 2025a; Drosos et al., 2025). While many systems are proposed for these GenUI applications, only a few benchmarks exist for evaluating model performance, and most of them focus on evaluating the design aspect of the UIs, either current UIs (Duan et al., 2024) or AI-generated UIs (Li et al., 2024; Si et al., 2024).

In this paper, we study a challenging GenUI application—dynamic widget generation for existing applications using the application's APIs. This problem is challenging for models because it requires both complex code and API reasoning as well as understanding the app context and intuitive user interface design. At the same time, it offers a unique insight into studying the functionality of the generated UIs in addition to their design, as we can design test cases to evaluate the functional correctness of the widgets similar to unit tests for traditional code tasks.

The widget generation task assumes an application with an existing UI and exposed APIs, and involves creating widgets—interactive UI elements displayed in a side panel—that allow users to input controls and trigger changes within the app through the exposed APIs. Figure 1 shows an example of a custom widget for Microsoft Excel—to hide/unhide columns. The widget generation problem involves automatically generating these widgets based on user's specification. As noted in Vaithilingam et al. (2024), AI-guided widget generation has the potential to make existing applications dynamic and personalized. By responding to user needs, AI-powered custom user interface generation can enable apps to adapt on the fly, provide easier discovery of complex actions, and support custom interfaces for tedious repetitive tasks.

Figure 1: An example widget to hide/unhide columns for an Excel document.

Generating dynamic widgets presents several challenges for foundation models, and requires advanced coding capabilities, intuitive interface design, and UI expertise. From the coding perspective, the task involves generalizing the logic for an action within the app to accommodate diverse inputs and application states; widgets must support internal state management to support memory and respond to (multiple) user interactions, implement event handlers to react to changes in the widget as well as the app, and use the app's APIs appropriately, as well as write responsive HTML and CSS layouts. On the design and UI side, creating intuitive user interfaces requires accurately inferring user intent from natural language utterance and the app context, providing user-friendly input methods (e.g., dropdowns, text boxes, cascading menus), ensuring ease of use through features like shortcuts and appropriate defaults, and maintaining logical layouts and flow that align with the users' mental model. Achieving these goals requires a combination of coding expertise, domain knowledge in UI/UX principles, and spatial reasoning.

To study the dynamic widget generation problem, we curate a set of evaluation benchmarks based on Microsoft Office apps (Excel and Word). We chose Office apps because they provide well-developed JavaScript (OfficeJS) APIs (especially for Excel and Word) that have been used extensively by developers to build third-party add-ins. Moreover, these apps are complex applications with many features that overwhelm their users, who frequently turn to resources like Stack Overflow to figure out how to perform their tasks. This makes Office a promising domain for leveraging LLMs to automatically generate dynamic personalized widgets from natural language utterance. It also serves as an important setting for studying foundation models' performance in real-world contexts, given the complexity of the apps and their APIs.

To create this benchmark, we scraped OfficeJS code samples from its official documentation and augmented them with AI-generated task descriptions and test scenarios. In addition to the Office-based dataset, we curate widget tasks for several synthetic domains to enable more controlled studies with limited APIs and easily verifiable application states using an existing semantic parsing dataset (Givoli & Reichart, 2019). These synthetic domains are designed to capture common interaction patterns–such as managing calendar events, navigating file systems, and sending messages and target some key data types such as dates and times, lists and recursive objects. Together, the Office-based and synthetic benchmarks provide a balanced suite for evaluating both the practical utility and the fundamental capabilities of foundation models for dynamic widget generation.

Finally, evaluating AI-generated widgets is challenging. Online evaluation would require users to actively interact with the widgets and see if it produces the desired outcomes, but this process is time-consuming and does not scale. To address this, we propose a new offline evaluation pipeline that executes the widget code in a real app environment to detect errors, uses LLM-generated Playwright scripts (Microsoft, 2020) to simulate user interactions, and runs test scripts to check for functional correctness of the widget along with using a multimodal model to assess design and usability.

In our evaluation, we compare popular models with various prompting and generation strategies. For Excel and Word domain, closed-source models such as GPT-4o, GPT-4.1, and o4-mini outperform open-source models such as LLaMa-3.1-70B, but even with access to tools for API documentation search and code compilation, their success rates on Excel and Word widget tasks remain around 50%. Models exhibit runtime errors, such as assuming the existence of certain objects in the document without checking first. They also make widget logic errors, for example, performing actions on the wrong objects. We also found that models generally require detailed task instructions to perform well on complex apps and are capable of generating widgets using both simple and MUI UI libraries. In synthetic domains, GPT-4o achieves over 95% success on two domains and around 80% on the third. LLaMa and Phi-4 struggle, particularly with handling dates and times. Across all domains, we also observed some UI design issues that could be improved, such as brittle input fields requiring long texts or exact font names, which can be difficult for users to fill correctly.

## 2 THE WIDGET GENERATION PROBLEM

### 2.1 WIDGETS

A widget is a custom interface that enable users to interact with the host application in a task-specific way. In our setup, a widget is a React Component that interfaces with the host application through the app's APIs. The widgets are rendered in a side-pane by the app as shown in Figure 1, accompanied by callback functions that invoke the necessary API functions. In this example, the widget reads column headers from the first row of a spreadsheet and allows the user to show or hide columns without directly editing the spreadsheet manually.

Widgets work by reading and modifying the host app's underlying state. This state—such as the spreadsheet context object in Excel, the document context object in Word, or a list of "Event" objects for the calendar app—is exposed to the widget using a pre-defined set of API functions. Note that the host app's state can also be modified externally, either by the user directly (e.g., editing cell values) or through other widgets. Therefore, for a widget to both be useful and function correctly, it needs to work well within the context of the host app, carefully make changes to the app's state, as well as be reactive to external changes to the app. To achieve this, usually widgets need to have one or many of the following complex features:

**Wrap controllable inputs around complex functionality.** The widgets need to provide input interfaces for triggering specific actions using the app's API. For example, a checkbox widget might allow the user to input a column letter and, based on the input, call the columnHidden API function to hide the specified column in Excel.

**Reactive to data changes.** This functionality goes beyond simple input wrappers by displaying dynamic content/personalizing widget UI dynamically based on data derived from the app's state. For instance, instead of a static text box for inputting a column letter, the widget could display checkboxes populated with column names extracted from the first row of the spreadsheet. Such widgets rely on the app's API to fetch relevant data and generate the appropriate user interface elements. They also need to remain synchronized with the app's state, as external changes (e.g., column renaming or reordering) can impact their functionality. This synchronization is typically achieved by using explicit event handlers into the application if the application's API provides one.

**Recording state.** Most often widgets need to maintain their own internal state to track user actions or store information for future operations. For example, a widget designed to support "undo" functionality or cascading inputs might store a history of user actions. These widgets require robust state management to ensure consistency and responsiveness.

**Example.** Figure 3 in the Appendix shows the widget code for the Hide/Unhide columns task.

### 2.2 THE WIDGET GENERATION PROBLEM

**Setup.** We consider the problem of generating task-specific widgets dynamically within existing applications such as Microsoft Excel or Word. The setup begins with an existing application state—for example, a particular document or spreadsheet—in which the user wishes to perform a task (such as hiding a particular column) that requires a custom interface. Rather than writing widget code manually, the user provides a natural language description of the desired functionality or of the widget itself.

**Input to the foundation models.** In addition to the user provided natural language description, we can automatically gather more relevant context for the widget generation problem. This includes the name of the host application, a summary of the current application state, and the application's APIs that the widget can use. In the case of Excel and Word, these APIs are the publicly documented OfficeJS interfaces that large language models are already pre-trained on. For synthetic applications used in our experiments (e.g., calendar, file, or messenger), we provide an explicit TypeScript interface describing the available operations.

**Access to tools.** In some experimental configurations, the LLM is given tool access in addition to the above contextual information. Specifically, for Excel and Word apps that have a very extensive API, we supply embeddings of each API call for retrieval-augmented generation (RAG), allowing

the model to ground its widget code more reliably. The model is also granted access to a TypeScript compiler to check for compile errors before returning a final answer.

**System prompt.** The system prompt provides additional guidance: it explains the setup, presents a high-level widget template, specifies output formatting, and outlines good widget design principles (e.g., the widget should be intuitive, user friendly, and reactive). It also highlights implementation practices to make automated evaluation of these widgets easier, such as assigning unique IDs to HTML elements to support automated testing and evaluation.

**Output.** The model output is a fully fleshed-out React TypeScript code for the widget component.

## 3 BENCHMARKS

To evaluate the widget generation capabilities of foundation models, we construct two sets of benchmarks: (a) widgets for real-world complex applications such as Microsoft Office apps, and (b) widgets for three synthetic application domains, namely Calendar, File System, and Messenger.

**Benchmark components.** Each benchmark task consists of the following components: a widget description and a set of test scenarios designed to evaluate the widget. Each test scenario includes: (i) an initial script that sets the host application to a predefined state, (ii) a detailed natural language description specifying the change the user wishes to make to the application, and (iii) a verification script that checks whether the application state after executing and interacting the widget matches the expected outcome for the scenario.

Some test scenarios are designed to be executed sequentially. In these cases, the user begins with an initial document, generates a widget, and performs the first test scenario. Subsequent scenarios operate on the modified application state produced by the previous scenario. In other cases, scenarios are independent: each scenario begins with a separate document, and the same widget is be reused across different documents to perform distinct tasks.

**Types of user utterances.** The exact natural language utterance a user provides to a model for widget-generation can take several forms. One option is for the user to provide the overall widget description corresponding to the entire task. Alternatively, the user may supply the task description for a single test scenario. This second form mirrors the type of input commonly used with current copilot systems, where a user issues a command such as "Hide the product column in my data" rather than requesting a generalized widget like "Create a widget to hide or unhide columns in my data." In either case, the LLM must anticipate additional tasks the user might want to perform with the widget to avoid generating overly specific widgets that would have limited usability. A third form of utterance is to include all task descriptions from all test scenarios. This provides the generation model with the most complete information for the benchmark task but imposes a high burden on the user. Studying this setting is still valuable, as it allows us to measure the best possible performance of the LLM. In practice, this burden on the user can be mitigated by having the AI system infer likely usage scenarios.

### 3.1 OFFICE WIDGETS BENCHMARK GENERATION

This dataset is a collection of 35 Excel tasks and 15 Word widget tasks. For these domains, we represent the initial document as the result of running a snippet of OfficeJS code (init_code) starting from an empty document. This allows us to easily reset the document to its initial state during evaluation by clearing the current content and re-executing the code. Similarly, to verify the final state of the app after widget execution, we include verification scripts (verif_script) in the dataset. Each script is a snippet of OfficeJS code that reads the relevant application state and compares it to the expected values for the task. Figure 4 and Figure 5 in the appendix show example Excel and Word widget test scenarios along with their init_code and the verif_script. For these domains, each widget task's test scenarios are expected to be executed sequentially as followups over multiple turns of widget usage. The figures also show NL description of the followup test scenario, and the screenshots of the expected document after the followup.

There is no existing dataset that contains all of these components. Therefore, we craft a synthetic data generation pipeline using LLMs and execution feedback with the following 5 steps:

**Extracting grounding contexts.** To bootstrap the LLM-based synthetic dataset generation for the widgets, we start with some grounding contexts. In this case, since LLMs are unreliable at generating OfficeJS code, we start from OfficeJS code snippets from their documentation. From the OfficeDev Github repository (OfficeDev, 2025), we collect 141 code snippets for Excel and 58 Code snippets for Word. Each code snippet has a brief description text, OfficeJS code to do an action in the Office document, and often a OfficeJS code snippet to setup a sample document.

**Synthetic task generation.** Given this grounding context from OfficeJS documentation snippets, we pass each snippet to an LLM (in our case, GPT-4o) to act as a teacher framing coding tasks for a student's assignment on teaching a class on OfficeJS APIs. The LLM is asked to formulate a commanding task (i.e. perform a certain specific action) and express it in unambiguous and clear fashion (see Figure 4 (a)). The LLM is also asked to generate a init_code to generate a sample document with the data relevant to the task (see Figure 4 (b)). Then, the LLM is asked to generate the gold solution task_code and a verif_script to automatically grade student's submitted solutions. Note that the gold task_code is actually not required for the dataset but it helps with the later stages of the pipeline such as LLM self-repair, filtering and manual refinement steps. Also note that the task_code is very different from the gold widget_code as the task_code is only doing the commanding task (only expected to work on the current document and current NL spec) whereas the widget_code needs to generalize to multiple documents and expose inputs to the user to perform multiple tasks.

**Self-repair using execution feedback.** LLM generated OfficeJS code can be wrong (have compile, runtime errors, or not accurately do the task) even in the presence of the grounding context. Therefore, as next step, we compile and execute the generated init_code, task_code, and verif_code in that order and collect any compile or runtime errors. Adding these error messages as feedback to the LLM, we perform multiple rounds of LLM self-repair to fix the errors.

**Filtering + manual task editing.** At the end of the self-repair step, we discard any tuple of init, task, and verification code snippets that still has compile/runtime errors. Then, we perform manual inspection to check if the task_code is correct, and check if verif_code correctly marks a correct task_code as success and a wrong task_code as fail. Screenshots of the documents as well as code inspection are used for this step. Finally, after all these steps, we collect a set of well-curated 35 Excel tasks and 15 Word tasks for our evaluation.

**Generating followup test scenario.** Finally, we curate a followup test scenario to test the widget's ability to perform more than one task. We again use an LLM (GPT-4o in this case) to generate followup task description, any updates to the document (mimicking updates that a user would do before the follow-up), gold task_code for the followup task and finally another verification code to check the success of the followup task. These generated code snippets also undergo the LLM self-refinement loop using execution feedback as well as minor manual editing. Generating and manual fixing of follow-up tasks is significantly more efficient than the original tasks as the follow-up tasks can reuse most of the code snippets from the original task.

## 3.2 SYNTHETIC DOMAINS BENCHMARKS

To augment the above benchmark suite, we also curate widget tasks for several synthetic domains to enable more controlled studies with limited APIs and easily verifiable application states. For this, we start with a semantic parsing dataset containing natural language commanding instructions (NLI) (Givoli & Reichart, 2019), which is corpus of user utterances for 7 simple synthetic application domains, out of which we select 3 target applications. Each example in the dataset is a tuple of a) the application's initial state, b) an natural language description for a task to be carried out in this application context, and c) the expected state of the application after the task is executed successfully (see example in Figure 6 in the Appendix). This dataset is initially designed as a semantic parsing benchmark to test an LLM's ability to generate code (given an app's API) to solve tasks specified in natural language. Below we see, how we can convert this dataset into a widget dataset.

**Converting NLI dataset to a Widget dataset.** To convert the NLI dataset into a widget dataset, we grouped similar NL specifications into one widget task. The idea is that there is one NL description for the widget task and each of the individual instances (task descriptions, initial and final app states) act as test scenarios for the widget's invocation on different initial states of the app and different inputs. Figure 7 in the Appendix shows an example of the group and the widget generated for this group using just the first NL utterances as the user query for the widget. We group similar

utterances using an LLM to extract an abstract structure of the user prompts and group them based on the abstract structure. Table 4 in the Appendix shows the total number of groups and average/range of number of individual tasks within each group for each of the 3 synthetic domains.

## 4 EVALUATION FRAMEWORK USING LLM DRIVEN UI SIMULATION

To evaluate an AI-generated widget for a task in the dataset, we perform both static and dynamic checks. We present the compilation and runtime setup of our evaluation framework in Appendix A.6.

**Simulating user interaction with the widget using PlayWright.** Once the widget is successfully loaded in the runtime environment, the ideal way to evaluate its functionality is by letting a person use it to complete the intended task. However, for scalable offline evaluation, we simulate user interactions using an LLM-generated PlayWright web execution script. This script drives the widget by programmatically setting or clearing inputs, clicking buttons, and performing other relevant actions based on the task specification. The PlayWright script is generated by a web executor agent LLM that has access to the widget code (which was also generated by an LLM and instructed to assign unique id attributes to all HTML elements), the task specification, and the verification script. Figure 8 in the Appendix shows an example of PlayWright web execution script generated by GPT-4o for the Calendar widget task shown in Figure 7.

To execute the PlayWright script, we load the app and the widget in a PlayWright WebDriver and dynamically run the generated script. Since these scripts are generated by an LLM, they can occasionally contain errors that lead to parsing failures—such as attempting to set the value of an input element with a nonexistent or incorrect ID. If no parsing errors occur, we then perform a validation check to ensure the script sets inputs correctly based on the task description using a multimodal model's vision reasoning. If we observe parsing errors or incorrect inputs, we regenerate the web execution script code based on the feedback and retry execution until there are no parsing or input errors. While the validation logic is not exhaustive, in our experiments, we find that it captures most common issues with the LLM generated web execution scripts.

**Evaluating functional correctness.** Finally, after simulating the user's actions on the widget, we check whether the widget successfully performed the intended task by executing the task's verification script. The verification checks both the presence and correctness of expected values—e.g., whether a chart was inserted using the correct data, or whether a range of cells was updated with the intended formatting. It also ensures that there are no unintended changes to the app's state.

**UI visual evaluation.** In addition to measuring the functional correctness of the widgets, we use a vision-model-based judge to score the widgets (on a 1-5 scale) across five key dimensions (inspired by UICrit's evaluation criteria (Duan et al., 2024)).
*1. Layout:* Assesses the spatial arrangement of elements, and that related items are visually coherent.
*2. Readability:* Evaluates text clarity, font size, contrast, and the clarity of labels.
*3. Usability:* Measures how easily users can interact with the widget, and the effort needed.
*4. Learnability:* Considers how quickly new users can understand the operation of the widget.
*5. Cognitive ease:* Evaluates how well the widget minimizes mental effort, ensuring that using the widget does not require significantly more effort than specifying the task in natural language.

## 5 EVALUATION

We evaluate various models, user utterance types, and UI libraries for the widget generation problem on both real application domains and synthetic applications. Our goal is to investigate the strengths of different models and to characterize the types of failures they exhibit in this setting.

**Baselines.** We compare several foundation models, including GPT-4o, GPT-4.1, o4-mini, Phi-4, and LLaMA-3.1-70B. For the Excel/Word domains which have complex APIs, we also consider an enhanced setting (`Tool calling`) that combines retrieval-augmented generation (RAG) for retrieving API documentation and a TypeScript compiler.

We also examine different forms of user utterances as inputs to the models. These include utterances derived from a single test scenario (`First task`), utterances that combine all test scenarios of a widget task (`All tasks`), and general widget descriptions (`Widget desc`). For UI libraries,

we let models generate widget code using both a simple React-based library (`Simple`) and a more feature-rich Material UI (`MUI`) library.

**Metrics.** We evaluate widgets using both quantitative and qualitative metrics. Quantitative metrics include **success rate**, which measures functional correctness across all test scenarios (`widget success rate`), on the first test scenario alone (`first test success rate`), and as an average across scenarios (`average test success rate`); **runtime error rate**, the average frequency of execution errors; **User simulation error rate**, detected through LLM-generated Playwright script execution and multimodal verification of widget input states; and **average usability score**, assigned by a multimodal model across five usability dimensions. Qualitative assessment involves categorizing functional errors (unsupported task, incorrect input handling, no output change, or incorrect output) and analyzing the usability issues of widgets produced by different models and prompt configurations. Each experiment is repeated three times, and we report the mean and standard deviation, with model responses generated at temperature 0.

## 5.1 Results on Office Widget Benchmarks

**Ablation on models.** We first compare various models given the most descriptive and unambiguous user utterance (`All tasks`) in Table 1. We see that models without the tool calling ability to retrieve API docs and use the TypeScript compiler exhibit high runtime error rates and consequently have lower widget success rates, with LLaMa performing the worst and o4-mini the best. Adding tool calling support reduces some runtime errors (particularly those from wrong API usage) and leads to some improvements in widget success rates. However, even the best performing model (o4-mini) only achieves a success rate of 54% on Excel tasks and 44% on Word tasks, showing that there is still substantial room for improvement in the foundation models' widget generation capabilities.

**Failure error types.** Figure 2 shows the distribution of failure error types of the generated widgets for the top-3 performing models. These errors are diagnosed using both runtime/verification error messages and LLM-based reasoning on the app screenshots before and after widget use. We can see that, most of the time, the model is able to generate a valid-looking UI for the widget (indicated by few load errors–which signifies that we are able to load these widgets and display them to the user), but widget interactions often trigger runtime error. These runtime errors are more predominant in Excel tasks than Word tasks; in Excel, the runtime errors typically arise because of invalid assumptions about existing structures (tables, charts, shapes) or incorrect API arguments (especially for formulas and comments). The Word domain has a few runtime errors due to improper APIs usage for elements like images or checkbox controls. We found some of these errors crash the widget at runtime, while others are caught by the widget and logged silently. In cases where the widget does not have runtime errors, we found that it fails to make any changes to the target document, either by not identifying the target object (e.g., a chart on a different sheet) or by producing undesired changes, reflecting logic errors or user intent misunderstanding.

We also found the number of errors due to LLM-based user simulation setup is very low for all models. Also, since the user instruction has detailed instructions from all test scenarios, we did not find many cases of the widget not being able to support a particular test scenario.

**UI design and usability.** The usability scores as judged by an multimodal judge are shown in Table 1 and the radar graphs for each of the 5-dimensions for some of the model comparisons is shown in Figure 10 (a) and (b). In general, tool-calling versions of the models produce slightly better UI designs, with minor differences among models, and widgets for the Word domain tend to be more usable than those for Excel domains. Excel tasks often require more inputs (for e.g. to specify chart properties), and widgets frequently require manual entry of row and column indices or long drop-downs, increasing cognitive burden and reducing usability.

Positive usability examples from widgets of both domains include effective use of dropdown menus or toggle checkboxes to select API parameters or task options, cascading input expansion based on previous selections, thoughtful status indicators and output displays within widgets, and undo options where prompted by the task. Negative examples include UI overload with too many or redundant controls, non-intuitive input types (e.g., hex codes, font names, indices), brittle inputs such as formulas that require exact formatting, missing labels, and design choices that increase cognitive load (e.g., requiring users to specify active worksheet manually).

| Model | Widget success rate (%) ↑ | Runtime error rate (%) ↓ | User simulation error rate (%) ↓ | Average usability score (1 to 5) ↑ |
|---|---|---|---|---|
| **Excel** | | | | |
| GPT-4o | $31 \pm 0$ | $55 \pm 4$ | $0 \pm 1$ | $\mathbf{3.52 \pm 0.06}$ |
| GPT-4.1 | $27 \pm 1$ | $36 \pm 2$ | $3 \pm 2$ | $3.31 \pm 0.12$ |
| o4-mini | $39 \pm 6$ | $49 \pm 4$ | $0 \pm 0$ | $3.37 \pm 0.05$ |
| LLaMa | $3 \pm 0$ | $89 \pm 0$ | $0 \pm 0$ | $3.34 \pm 0.07$ |
| GPT-4o tool calling | $41 \pm 3$ | $30 \pm 8$ | $2 \pm 1$ | $\mathbf{3.47 \pm 0.08}$ |
| GPT4.1 tool calling | $47 \pm 7$ | $26 \pm 3$ | $2 \pm 1$ | $\mathbf{3.46 \pm 0.11}$ |
| o4-mini tool calling | $\mathbf{54 \pm 2}$ | $\mathbf{19 \pm 3}$ | $2 \pm 1$ | $\mathbf{3.45 \pm 0.07}$ |
| **Word** | | | | |
| GPT-4o | $11 \pm 6$ | $41 \pm 4$ | $0 \pm 0$ | $\mathbf{3.68 \pm 0.08}$ |
| GPT-4.1 | $18 \pm 6$ | $48 \pm 6$ | $4 \pm 2$ | $3.20 \pm 0.13$ |
| o4-mini | $16 \pm 8$ | $43 \pm 10$ | $0 \pm 0$ | $3.47 \pm 0.13$ |
| LlaMa | $13 \pm 0$ | $70 \pm 4$ | $0 \pm 0$ | $3.41 \pm 0.15$ |
| GPT-4o tool calling | $38 \pm 3$ | $11 \pm 4$ | $0 \pm 0$ | $\mathbf{3.71 \pm 0.13}$ |
| GPT4.1 tool calling | $27 \pm 5$ | $14 \pm 6$ | $6 \pm 2$ | $\mathbf{3.64 \pm 0.13}$ |
| o4-mini tool calling | $\mathbf{44 \pm 3}$ | $\mathbf{8 \pm 7}$ | $0 \pm 0$ | $\mathbf{3.64 \pm 0.09}$ |

Table 1: Results on Excel/Word domains - Various models with and without tool calling ability to do RAG on OfficeJS documentation. Uses `Simple` UI library and `All tasks` instruction.

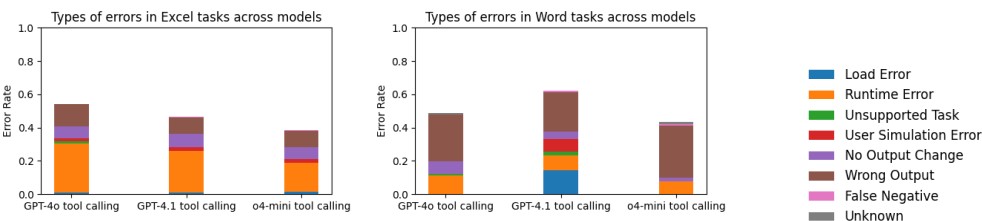

Figure 2: Types of errors for Excel and Word tasks across the top-3 performing models using Simple UI library and `All tasks` user instruction type.

**Ablation on user instruction type.** We compare different types of user instructions in Table 2; failure error types are shown in Figure 9, and the radar chart of usability scores is shown in Figure 11 (a) in the Appendix. Clearly, `All tasks` achieves the highest widget success rate and average test success rate. The `First task` instruction type performs better on the first test scenario for each widget but often fails to generalize to follow-up scenarios. `Widget desc` has the lowest success rate: while it produces valid widgets, they are often insufficient to complete the task due to the LLM's difficulty in anticipating user needs across the complex API surfaces of Excel and Word, as reflected in the higher rate of unsupported task errors (Figure 9). Figure 13 illustrates how a widget changes with varying verbosity levels of `widget desc` for a Word task. These results shows that widget generation can benefit from detailed instructions and from a setup where incremental, interactive follow-ups from the user could help guide the widget generation more effectively.

With respect to usability scores (Figure 11 (a)), widgets generated with `All tasks` instruction type are often better rated, especially along cognitive ease, learnability, and usability. This is because the model knows how the widgets will be used and can design an interface that minimizes user burden for the specified tasks.

**Ablation on UI library.** Finally, we test the widget generation problem when models are asked to generate widgets using either a simple React-based UI library or a more feature-rich Material UI library (Table 5 and Figure 11 (b)). We found larger models such as GPT-4o to perform well on both libraries, with `Simple` library setup performing slightly better and having better usability scores.

## 5.2 RESULTS ON SYNTHETIC DOMAINS

**Ablation across models.** Table 3 shows the widget success rates of GPT-4o, LlaMa-3.1-70B, and Phi-4. This experiment uses the `first task` as the user instruction and prompts the models to generate UI using simple libraries. Because this domain is simpler than Excel/Word domains in terms of API and task complexity, we found larger models such GPT-4o achieve near perfect performance (GPT-4o has ~95% performance on Calendar and Messenger and about 80% performance on

| Instruction type | Widget success rate | first test success rate | average test success rate | Runtime error rate | User simulation error | Average usability score |
|---|---|---|---|---|---|---|
| **Excel** | | | | | | |
| First task | $22 \pm 5$ | $\mathbf{61 \pm 4}$ | $42 \pm 4$ | $23 \pm 5$ | $3 \pm 1$ | $3.06 \pm 0.08$ |
| All tasks | $\mathbf{41 \pm 3}$ | $46 \pm 2$ | $\mathbf{46 \pm 1}$ | $30 \pm 8$ | $2 \pm 1$ | $\mathbf{3.47 \pm 0.08}$ |
| Widget desc | $21 \pm 6$ | $24 \pm 7$ | $23 \pm 7$ | $34 \pm 1$ | $1 \pm 2$ | $3.27 \pm 0.09$ |
| **Word** | | | | | | |
| First task | $9 \pm 3$ | $51 \pm 6$ | $31 \pm 3$ | $13 \pm 7$ | $1 \pm 2$ | $3.05 \pm 0.10$ |
| All tasks | $\mathbf{38 \pm 3}$ | $\mathbf{60 \pm 5}$ | $\mathbf{51 \pm 4}$ | $11 \pm 4$ | $0 \pm 0$ | $\mathbf{3.71 \pm 0.13}$ |
| Widget desc | $16 \pm 3$ | $22 \pm 3$ | $20 \pm 5$ | $16 \pm 8$ | $4 \pm 2$ | $3.25 \pm 0.07$ |

Table 2: Results on Excel/Word domains - GPT-4o tool calling with simple UI library for various types of user instruction types

| Model | UI library | Widget success rate (%) ↑ | Runtime error rate (%) ↓ | User simulation error rate (%) ↓ | Average usability score (1 to 5) ↑ |
|---|---|---|---|---|---|
| **Calendar** | | | | | |
| GPT-4o | Simple | $\mathbf{98 \pm 1}$ | $0 \pm 0$ | $0 \pm 0$ | $\mathbf{3.21 \pm 0.05}$ |
| LLaMa | Simple | $59 \pm 0$ | $0 \pm 0$ | $0 \pm 0$ | $3.11 \pm 0.06$ |
| Phi4 | Simple | $44 \pm 0$ | $2 \pm 0$ | $1 \pm 1$ | $3.09 \pm 0.06$ |
| **File System** | | | | | |
| GPT-4o | Simple | $\mathbf{81 \pm 3}$ | $0 \pm 0$ | $0 \pm 0$ | $\mathbf{3.21 \pm 0.08}$ |
| LLaMa | Simple | $77 \pm 0$ | $0 \pm 0$ | $5 \pm 2$ | $\mathbf{3.22 \pm 0.06}$ |
| Phi4 | Simple | $60 \pm 2$ | $0 \pm 0$ | $6 \pm 1$ | $\mathbf{3.20 \pm 0.08}$ |
| **Messenger** | | | | | |
| GPT-4o | Simple | $\mathbf{95 \pm 2}$ | $0 \pm 0$ | $1 \pm 1$ | $\mathbf{3.00 \pm 0.08}$ |
| LLaMa | Simple | $76 \pm 2$ | $1 \pm 1$ | $1 \pm 1$ | $2.81 \pm 0.04$ |
| Phi4 | Simple | $85 \pm 0$ | $0 \pm 0$ | $5 \pm 1$ | $2.85 \pm 0.04$ |

Table 3: Results on synthetic domains for various models using `first task` instruction.

Phi4). LLaMa and Phi4 seemed to perform poorly than GPT-4o with LLaMa beating Phi4 on 2 out of 3 domains (Calendar and File).

**Failure error types.** Figure 14 shows the distribution of failure errors types. We find all models have a low runtime error rate, because of the simpler API surface of the domains. However, LLaMa and Phi4 models have significant cases where the widget either fails to produce any change or generates wrong output (especially for the Calendar domain due to mishandling of date, time, and timezones, often resulting in empty filtered items). Some errors stem from intent misunderstanding—for e.g., when a user intends to delete two specific events, the widget deletes a range of items.

**Generalizability across test scenarios.** The widgets for these domains generally generalize well to other test scenarios for the same widget (see Table 6), because of grouping similar semantic parsing tasks together and the limited API surface. Phi4 shows slightly worse generalization, with some unsupported task failures (Figure 14). Some cases of failed generalization arose when the first task targets a specific object (e.g., the largest file), while subsequent tasks target a different object (e.g., the smallest file).

**Design and usablity.** Figure 15 shows the UI scores radar charts. GPT-4o achieves slightly higher usability score compared to other models (for Calendar and Messenger, in terms of layout, usability, cognitive ease). For example, in the Calendar domain, we found LLaMa and Phi4 widgets increase cognitive burden by requiring users to map temporal references such as (e.g. "Thursday" or "10th") to exact date/time pickers. Additionally, while GPT-4o widgets minimizes typing with dropdowns wherever possible, LLaMa and Phi4 widgets tend to rely on text boxes, reducing usability. Across domains, Calendar and File widgets generated by GPT-4o are easier to use and learn than Messenger widgets. Messenger widgets often had long lists of checkboxes and manual inputs for groups, which increases cognitive load for users.

**Simple vs MUI libraries.** GPT-4o and LLaMA achieve similar success rates across both UI libraries, with slightly better performance using the simple library (Tables 7 and 8). In contrast, Phi4 performs poorly with MUI (Table 9). For calendar domain, in particular, Phi4 has a higher runtime error rate, because it tries to use enums in the type definition to create widget labels; however, types are unavailable at runtime. In terms of usability, scores are generally comparable across libraries for GPT-4o, with one exception: for the File domain, MUI widgets have a higher score.

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

# A APPENDIX

## A.1 DATASET AND CODE RELEASE

We will publicly release the full dataset, evaluation framework code, and experimental data for the final publication, following an ethics and legal review by our institution.

## A.2 ADDITIONAL RELATED WORKS.

Recent work has explored using AI to generate user interfaces from high-level textual descriptions–creating UI mock-ups or graphic layouts from text (Huang et al., 2021; Lin et al., 2023a;b), designing menus (Kargaran et al., 2023), mobile UX layouts (Lu et al., 2023), flows across multiple screens (Lu et al., 2024a), interactive webpages (Xiao et al., 2024), and generating UI code from sketches (Li et al., 2024) and designs (Si et al., 2024). In contrast to these works, our focus is on generating smaller, task-specific widgets rather than full applications, and in scenarios where a user requests a widget on-the-fly rather than a developer using AI to assist with web design.

Dynamic UI generation similar to our work is also seen in previous works such as DynaVis (Vaithilingam et al., 2024), dynamic prompt middleware (Drosos et al., 2025), and generative interfaces (Chen et al., 2025a). Our work is closely related to and inspired by DynaVis, but rather than focusing on a single application of Vegalite-based visualization editing, we target complex real-world applications such as Excel and Word and establish a benchmark for evaluation. In contrast, the latter two works focus more on responding to user queries to LLMs by proactively generating user interfaces, which does not have the API integration aspect into an existing application like ours.

Computer use agents (CUA) (OpenAI, 2025b) and web automation agents (Mozannar et al., 2025) provide an orthogonal approach for having AI perform tasks in complex applications. Unlike web automation, widget generation leverages app APIs for faster execution and requires only a single LLM call to create reusable widgets. Moreover, our automatic evaluation framework itself uses web automation via Playwright to simulate user interactions with the generated widgets.

## A.3 RUNNING EXAMPLE'S WIDGET CODE

(a) Widget high-level template

```
export const
    ShowHideColumnsWidget:React.FC<WidgetProps> =
  ({ /* Data about the host app */ }) => {
  /* Declare widget's state */
   /* React to data changes */
   /* Callbacks to handle UI (encapsulate
       app's APIs) */
return (
  /* UI components */
  );};
```

(b) Data about the host app. Computed once in the parent component and passed as a property to all widgets

```
const [data, setData] = useSharedState();
useEffect (() => { Excel.run(async (context) => {
    sheets .onChanged.add(handleSheetChange); // add
        listener
}); }, []);
const handleSheetChange = async (eventArgs) => { ...
    setData(range.values);  };
```

(c) Declare widget's state

```
interface UndoAction {
  index: number;
  prevHidden: boolean;
}

// State for column names, their hidden
    states, and undo history
const [columnNames, setColumnNames] =
    useState<string[]>([]);
const [columnHiddenStates, setColumnHiddenStates] =
    useState<boolean[]>([]);
const [undoStack, setUndoStack] =
    useState<UndoAction[]>([]);
```

(d) React to data changes

```
useEffect (() => { ...
  const headers = data[0] || []; // get column names
      from speadsheet
  setColumnNames(headers); setUndoStack([]); // update
      state
  Excel .run(async (context) => { ...
      const states = ranges.map(r => !!r.columnHidden);
          // get current status of columns
      setColumnHiddenStates(states); // update state
  });
}, [data]); // Update state whenever the data
    variable changes
```

(e) UI callbacks logic

```
const setColumnHidden = async (index, hidden) => {
  Excel .run(async (context) => { ...
      colRange.columnHidden = hidden; }); // call
      relevant Excel API
};
const handleCheckboxToggle = (index) => {
  setUndoStack ([... undoStack, { index, prevHidden }]);
      // Update undo stack for later
  setColumnHidden(index, !prevHidden);
      setColumnHiddenStates(...); // Trigger
      changes in the app
};
const handleUndo = () => { ...
      setColumnHidden(last.index, last .prevHidden); };
      // Trigger changes in the app
```

(f) UI components

```
<div id="columnVisibilityWidget">
  ...
  {columnNames.map((name, idx) => (
    <label><input type="checkbox" checked={...}
      onChange={() => handleCheckboxToggle(idx)} />
          {name}</label>
  ))} {/* Map column names to checkboxes */}
  <button disabled={undoStack.length===0}
      onClick={handleUndo}>Undo</button> {/*
      Button for undo */}
</div>
```

Figure 3: Widget code for hide/unhide checkboxes task in Figure 1; Code is broken down into various sub-components.

## A.4 Office widget task examples

(a) Detailed description of the first test scenario

Format the table with the following specifications : Set the header row background color to blue (#0000FF), the data rows background color to light gray (#D3D3D3), the second column background color to yellow (#FFFF00), and the first row (excluding headers) background color to light green (#90EE90).

---

(b) init_code to create a sample Excel sheet

```
await Excel.run(async (context) => {
  const sheet =
      context.workbook.worksheets.add("Sample");
  const table = sheet.tables.add("A1:D1", true /*
      hasHeaders */);
  table.getHeaderRowRange().values = [["Date",
      "Merchant", "Category", "Amount"]];
  table.rows.add(null, [ ["1/1/2017", "The Phone
      Company", ...], ... ]); // sample rows
  sheet.activate(); await context.sync(); });
```

(c) Screenshot of the initial document

---

(d) verif_script to verify functional correctness

```
await Excel.run(async (context) => {
  ...
  const headerRange = table.getHeaderRowRange();
  headerRange.load("format/fill/color");
  await context.sync();
  assert(headerRange.format.fill.color === "#0000FF",
      "Expected header row color #0000FF");
  // Similar checks for data rows, column B,
      and first row
});
```

(e) Screenshot of the expected document

---

(f) Detailed description of the followup test scenario

Change the header row background color to green (#008000), and the last row background color (excluding headers) to light coral (#F08080). Leave all other formatting as is .

(g) Screenshot of the expected document

Figure 4: Test scenarios for a table format editing widget task for Excel along with a detailed NL description, init_code, verif_script, and the screenshots for the initial document and the expected final document.

(a) Detailed description of the first test scenario

Perform a basic search for the word 'functionality' **in** the document and highlight all occurrences with a blue background color.

---

(b) init_code to create a sample Word document

```
await Word.run(async (context) => {
  const body = context.document.body;
  body.clear();
  body.insertParagraph("Do_you_want_to_create_a_
      solution_that_extends_Word?", "Start");
      // ...
  body.paragraphs.getLast().insertText("Use_add-in_
      commands_to_extend_the_UI_and_run_
      JS...", "Replace"); // ...
  await context.sync();
});
```

(c) Screenshot of the initial document

(d) verif_script to verify functional correctness

```
await Word.run(async (context) => {
  // Verify 'functionality' is highlighted
      in blue
  const results =
      context.document.body.search("functionality");
      ...
  results.items.forEach(i =>
      i.font.load("highlightColor")); ...
  assert(results.items.every(i => i.font.highlightColor
      === "#0000FF"), "...");

  // Ensure no unnecessary changes
  const paras = context.document.body.paragraphs;
      paras.load("items"); ...
  paras.items.forEach(p =>
      p.font.load("highlightColor")); ...
  assert(paras.items.every(p => p.font.highlightColor
      === "" || results.items.includes(p)), "...");

  console.log("Verification_completed_
      successfully");
});
```

(e) Screenshot of the expected document

(f) Detailed description of the followup test scenario

Undo the previous highlight and instead perform a basic search for the word 'add-in' **in** the document and highlight all occurrences with a yellow background color ..

(g) Screenshot of the expected document

Figure 5: Test scenarios for word search and highlight task for Word along with NL utterance, init_code, verif_script, and the screenshots for the initial document and the expected final document.

### A.5 SYNTHETIC DOMAIN EXAMPLES

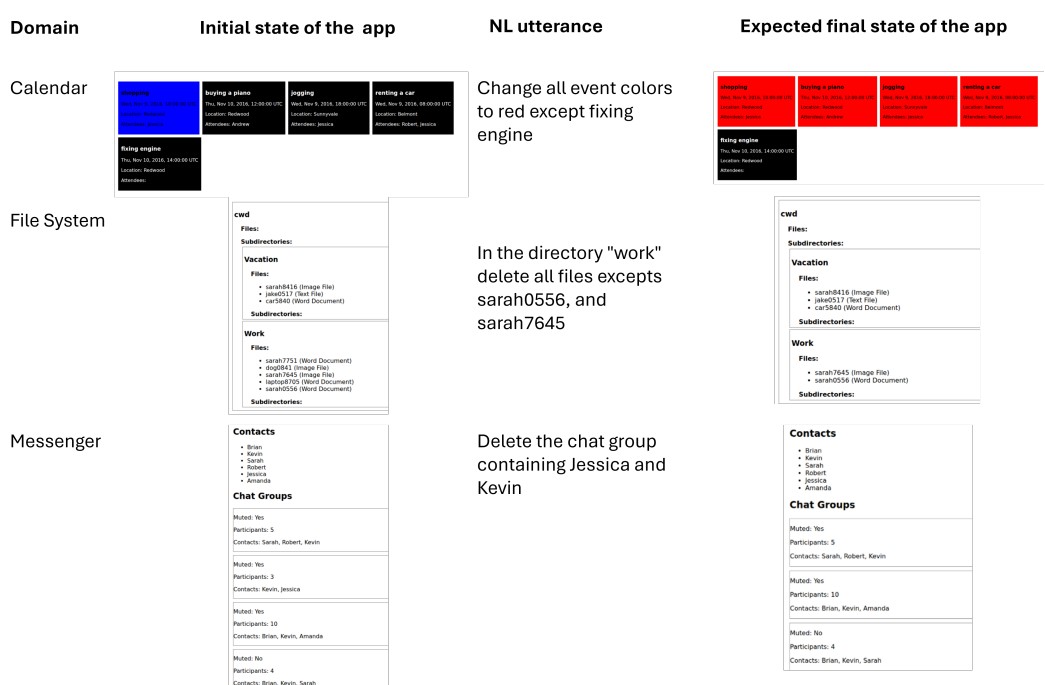

Figure 6: NLI dataset examples for the 3 domains – Calendar, File System, and Messenger. Each task consists of the initial state of the app, the NL utterance to specify a desired action, and the exepcted final state of the app.

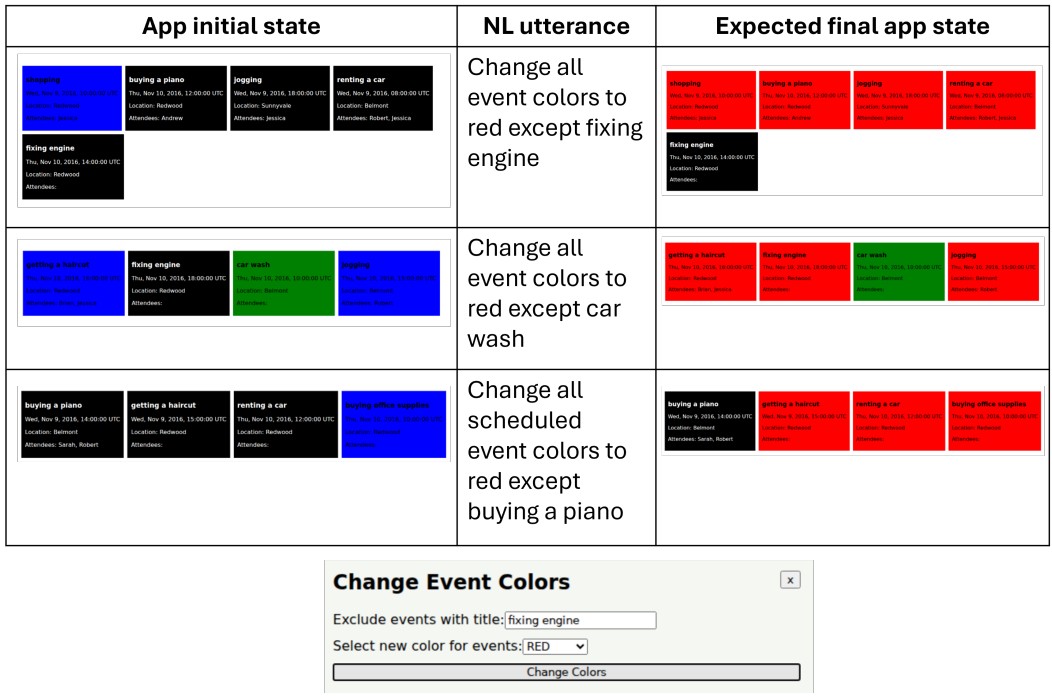

Figure 7: Example of a group in the Calendar domain along with the widget generated by GPT-4o for this group of tasks.

| Domain | Total # of tasks | Average test scenarios per task | Min test scenarios per task | Max test scenarios per task |
|---|---|---|---|---|
| Calendar | 34 | 1.8 | 1 | 3 |
| File system | 26 | 1.8 | 1 | 3 |
| Messenger | 26 | 2.0 | 1 | 3 |

Table 4: Statistics for synthetic domains widgets datasets for the three domains – calendar, file system and messenger

### A.6 EVALUATION FRAMEWORK DETAILS

**Compiling widget code**   We use the TypeScript compiler for this step, configured with type definitions for the synthetic app's internal APIs and the OfficeJS library. This compilation checks for basic correctness, including syntax errors, type mismatches, and misuse of known APIs. However, it cannot catch all possible errors—particularly those that arise due to OfficeJS's asynchronous execution model or application-specific runtime context. Such issues are surfaced later during widget execution in a fully instantiated app environment. Some errors detected at this stage include:

- Argument of type '"ClusteredColumn"' is not assignable to parameter of type '"ColumnClustered"' — ...'. This often occurs when the LLM generates a chart type name in a human-readable format ("ClusteredColumn") instead of using the exact expected string literal ("ColumnClustered").

- Property 'comments' does not exist on type 'Range'. This reflects a misunderstanding of the API's object model, where features like comments must be accessed through different APIs (e.g., worksheet.comments) rather than directly from the range object.

- Property 'values' does not exist on type 'TableCell'. This highlights a common LLM mistake where it incorrectly pluralizes or generalizes property names based on intuition rather than the actual API definition.

**Runtime verification on loading the widget.**   Once the widget code compiles successfully, we verify its behavior in a live application environment. For Office apps, we interface with the live application through their add-in infrastructure. And we orchestrate something similar for the synthetic domains as well. We create a dummy addin that can receive the generated widget code over a socket connection and dynamically executes the code to render the widget component. We refer to this process as *loading* the widget.

Before loading, we first initialize the application with the appropriate starting state defined by the task. This is done by sending the init_code—also transmitted over the socket—which sets up the document with the expected input content and structure.

On loading the widget, there can be several kinds of runtime errors. These include HTML-related issues, such as malformed elements or conflicting DOM IDs, as well as API errors (especially OfficeJS API) caused by the widget attempting to access or manipulate the app's state during initialization. For example, many widgets attempt to populate dropdowns, checkboxes, or default values by querying the document state; if these queries do not sync the context properly, read values before loading in OfficeJS, or assume non-existent objects, they can result in runtime exceptions.

For example, for one of the Calendar tasks, GPT-4o creates a widget code that has following snippet:

```
<option key={event.title} value={event.title}>
    {event.title} @ {event.startTime} ({event.location})
</option>
```

This throws a runtime error on loading the widget as "Objects are not valid as a React child (found: [object Date])" as event.startTime is a Date object and GPT-4o is injecting it directly into the JSX, which causes this error.

```
# Task spec: Change all event colors to red except fixing engine
# set appropriate inputs after first clearing any previous inputs

# Clear and set the "Exclude events with title" input
exclude_title_input = page.locator('# exclude_title_input')
await exclude_title_input.press('Control+a')
await exclude_title_input.press('Delete')
expected_exclude_title = 'fixing-engine'
await exclude_title_input.fill(expected_exclude_title)
read_exclude_title = await exclude_title_input.input_value()
check_inputs_logs.append((expected_exclude_title, read_exclude_title,
    expected_exclude_title.lower() == read_exclude_title.lower()))

# Set the "Select new color for events" dropdown
color_picker = page.locator('#color_picker')
await color_picker.select_option('RED')
read_color = await color_picker.input_value()
check_inputs_logs.append(('RED', read_color, 'red' == read_color.lower()))

# Click the "Change Colors" button
await page.locator('#change_colors_button').click()
```

Figure 8: PlayWright web execution script generated by GPT-4o for the Calendar widget task in Figure 7

### A.7 ADDITIONAL EXCEL AND WORD DOMAIN RESULTS

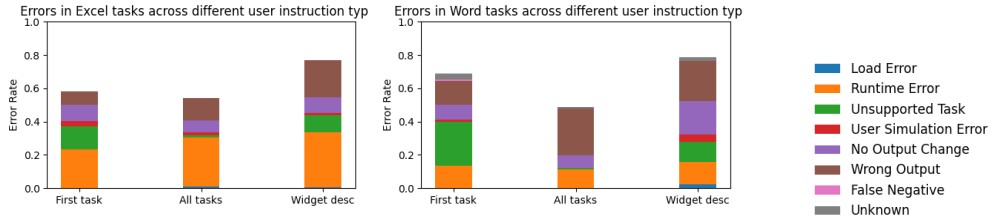

Figure 9: Types of errors for Excel and Word tasks for various user instruction types using GPT-4o tool calling variant and Simple UI library.

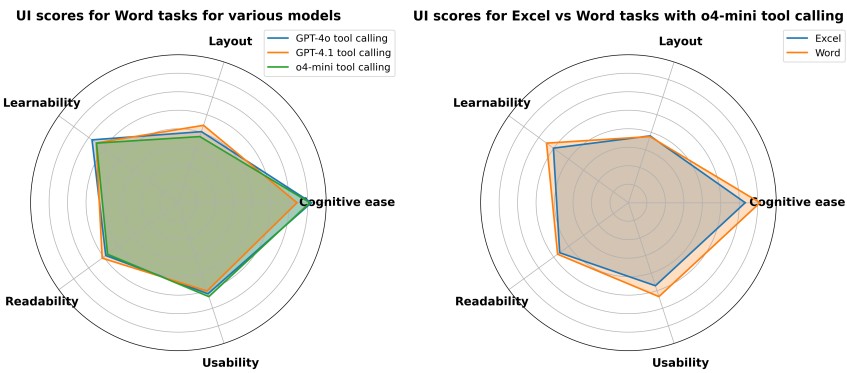

Figure 10: Comparison of UI scores across different models and domains for Excel and Word.

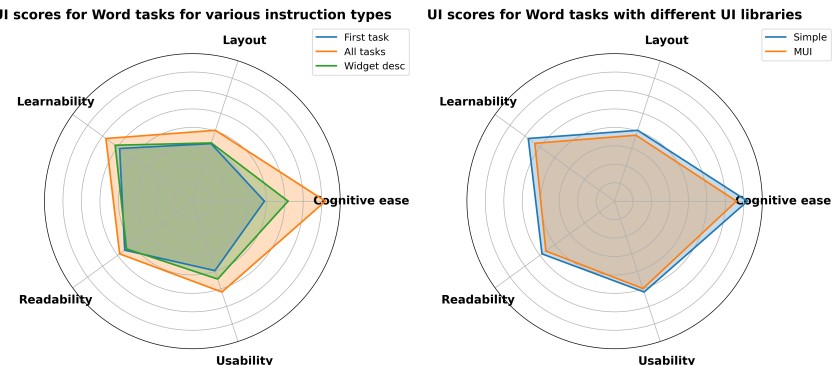

Figure 11: Comparison of UI scores for various instruction types and UI libraries for Excel and Word using GPT-4o tool calling.

| UI library | Widget success rate | Runtime error rate | User simulation error rate | Average usability score |
|---|---|---|---|---|
| **Excel** | | | | |
| Simple | **41 ± 3** | 30 ± 8 | 2 ± 1 | **3.47 ± 0.08** |
| MUI | **40 ± 2** | 30 ± 2 | 2 ± 1 | 3.35 ± 0.06 |
| **Word** | | | | |
| Simple | **38 ± 3** | 11 ± 4 | 0 ± 0 | **3.71 ± 0.13** |
| MUI | 33 ± 5 | 13 ± 9 | 1 ± 2 | 3.53 ± 0.15 |

Table 5: Results on Excel/Word domains - GPT-4o tool calling with all tasks instruction type for various types of UI libraries (Simple vs MUI)

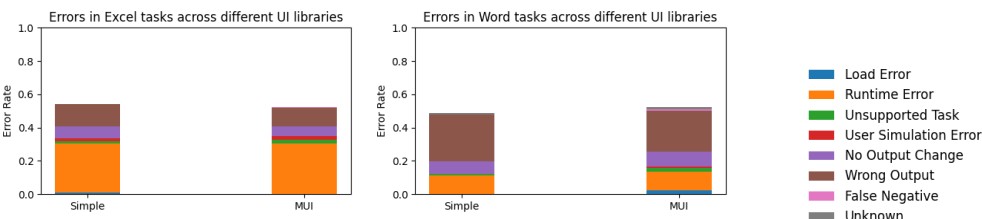

Figure 12: Types of errors for Excel and Word tasks for various UI libraries using GPT-4o tool calling variant and `All tasks` user instruction type.

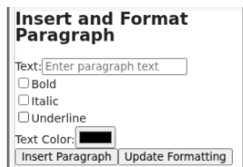 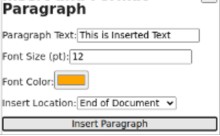 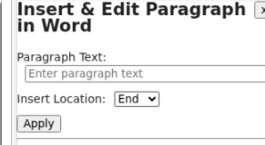

V2: Create a widget to insert a new paragraph with specific text, apply orange highlight and bold formatting, and later modify it to change the highlight to yellow and remove bold formatting, updating the document.

V3: Create a widget to insert a paragraph with text, apply formatting, and modify the formatting later, updating the document.

V4: Create a widget to insert and modify a paragraph with specific formatting in a Word document.

V5: Create a widget to insert and edit a paragraph in Word.

Figure 13: Widgets generated for a Word task using `Widget desc` instruction type of various verbosity levels.

## A.8    ADDITIONAL SYNTHETIC DOMAIN RESULTS

| Model | UI library | Widget success rate | First test success rate | Average test success rate |
|---|---|---|---|---|
| **Calendar** | | | | |
| GPT-4o | Simple | $98 \pm 1$ | $99 \pm 1$ | $99 \pm 1$ |
| LLama | Simple | $59 \pm 0$ | $59 \pm 0$ | $54 \pm 0$ |
| Phi4 | Simple | $44 \pm 0$ | $47 \pm 0$ | $43 \pm 0$ |
| **File System** | | | | |
| GPT-4o | Simple | $81 \pm 3$ | $88 \pm 3$ | $90 \pm 2$ |
| Llama | Simple | $77 \pm 0$ | $85 \pm 0$ | $80 \pm 1$ |
| Phi4 | Simple | $60 \pm 2$ | $68 \pm 2$ | $65 \pm 1$ |
| **Messenger** | | | | |
| GPT-4o | Simple | $95 \pm 2$ | $97 \pm 2$ | $95 \pm 2$ |
| LLama | Simple | $76 \pm 2$ | $81 \pm 3$ | $76 \pm 2$ |
| Phi4 | Simple | $85 \pm 0$ | $96 \pm 0$ | $88 \pm 0$ |

Table 6: Results on synthetic domains - widget success rate vs first test success rate vs average test success rate using `First task` instruction.

| Model | UI library | Widget success rate | Runtime error rate | User simulation error rate | Average usability score |
|---|---|---|---|---|---|
| | | | **Calendar** | | |
| GPT-4o | Simple | $\mathbf{98 \pm 1}$ | $0 \pm 0$ | $0 \pm 0$ | $\mathbf{3.21 \pm 0.05}$ |
| GPT-4o | MUI | $92 \pm 1$ | $0 \pm 0$ | $2 \pm 1$ | $\mathbf{3.21 \pm 0.07}$ |
| | | | **File System** | | |
| GPT-4o | Simple | $\mathbf{81 \pm 3}$ | $0 \pm 0$ | $0 \pm 0$ | $3.21 \pm 0.08$ |
| GPT-4o | MUI | $\mathbf{82 \pm 5}$ | $0 \pm 0$ | $1 \pm 2$ | $\mathbf{3.48 \pm 0.05}$ |
| | | | **Messenger** | | |
| GPT-4o | Simple | $\mathbf{95 \pm 2}$ | $0 \pm 0$ | $1 \pm 1$ | $\mathbf{3.00 \pm 0.08}$ |
| GPT-4o | MUI | $79 \pm 5$ | $1 \pm 2$ | $3 \pm 2$ | $\mathbf{2.93 \pm 0.14}$ |

Table 7: Results on synthetic domains - GPT-4o simple UI library vs MUI

| Model | UI library | Widget success rate | Runtime error rate | User simulation error rate | Average usability score |
|---|---|---|---|---|---|
| | | | **Calendar** | | |
| LLaMa | Simple | **59 ± 0** | 0 ± 0 | 0 ± 0 | **3.11 ± 0.06** |
| LLaMa | MUI | **58 ± 1** | 2 ± 0 | 3 ± 1 | 2.97 ± 0.04 |
| | | | **File System** | | |
| LlaMa | Simple | **77 ± 0** | 0 ± 0 | 5 ± 2 | **3.22 ± 0.06** |
| LLaMa | MUI | 73 ± 0 | 16 ± 1 | 4 ± 0 | 3.05 ± 0.09 |
| | | | **Messenger** | | |
| LLaMa | Simple | **76 ± 2** | 1 ± 1 | 1 ± 1 | **2.81 ± 0.04** |
| LLaMa | MUI | 73 ± 5 | 1 ± 1 | 8 ± 2 | **2.84 ± 0.08** |

Table 8: Results on synthetic domains: LLaMa simple vs MUI

| Model | UI library | Widget success rate | Runtime error rate | User simulation error rate | Average usability score |
|---|---|---|---|---|---|
| | | | **Calendar** | | |
| Phi4 | Simple | **44 ± 0** | 2 ± 0 | 1 ± 1 | **3.09 ± 0.06** |
| Phi4 | MUI | 23 ± 1 | 36 ± 0 | 5 ± 3 | **3.09 ± 0.08** |
| | | | **File System** | | |
| Phi4 | Simple | **60 ± 2** | 0 ± 0 | 6 ± 1 | **3.20 ± 0.08** |
| Phi4 | MUI | 56 ± 2 | 7 ± 1 | 8 ± 0 | **3.20 ± 0.07** |
| | | | **Messenger** | | |
| Phi4 | Simple | **85 ± 0** | 0 ± 0 | 5 ± 1 | **2.85 ± 0.04** |
| Phi4 | MUI | 79 ± 2 | 0 ± 0 | 1 ± 1 | **2.86 ± 0.05** |

Table 9: Results on synthetic domains - Phi4 Simple UI library vs MUI

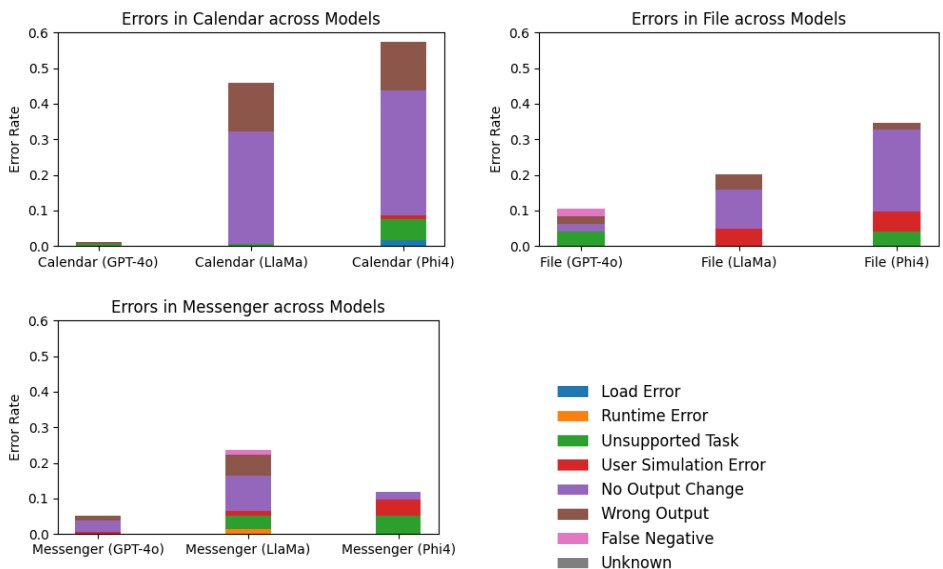

Figure 14: Types of errors for Calendar, File, Messenger for various models with `Simple` UI library and `first task` instruction.

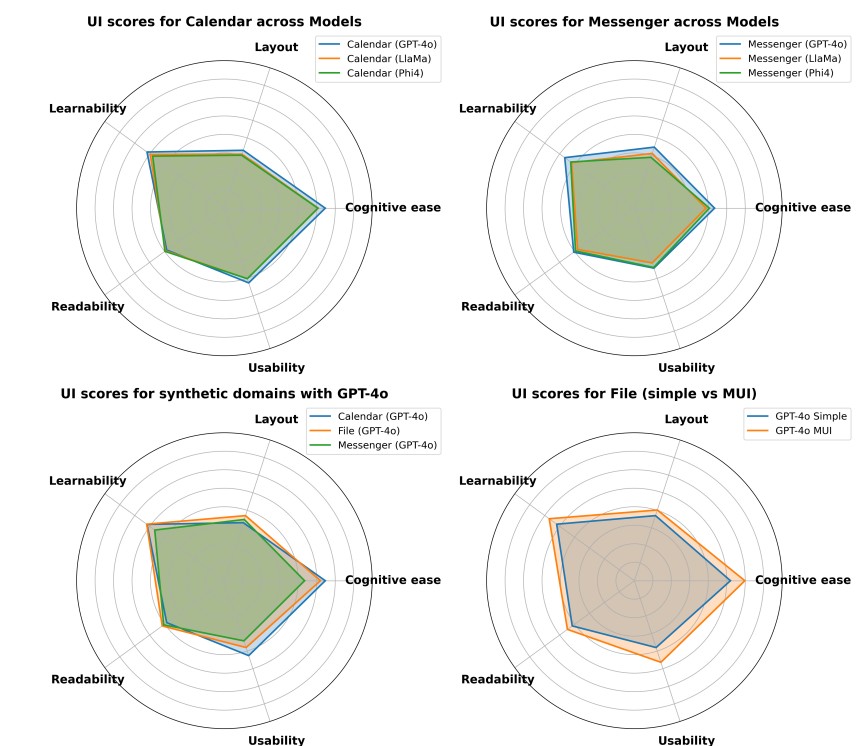

Figure 15: Comparison of UI scores on synthetic domains across different models and settings.

