# OpenReview forum: "WidgetEval: Benchmarking Foundation Models on Dynamic Widget Generation for Apps"
_ICLR.cc/2026/Conference — Submitted to ICLR 2026_

### Official Review · Reviewer_1cY8 · 2025-10-26

**Soundness:** 3
**Presentation:** 3
**Contribution:** 3
**Rating:** 6
**Confidence:** 3

**Summary:**

The paper introduces WidgetEval, a benchmark for assessing large language models (LLMs) on dynamic widget generation inside existing applications. The goal is to evaluate whether models can understand a user’s natural-language intent, the host app context, and its APIs to generate runnable React components that encapsulate app functionality (e.g., Microsoft Excel/Word via OfficeJS). The work primarily aims to (i) draw attention to a new, practically relevant evaluation setting and (ii) contribute an evaluation pipeline and dataset (tasks, test scenarios, verification scripts) to study model capabilities and failure modes.

**Strengths:**

The data construction process is well-designed and fully traceable, involving a rigorous multi-stage pipeline that spans from OfficeJS documentation crawling and LLM-based synthesis to execution feedback–driven self-repair and manual verification. This design, including a self-repair loop, reflects a strong commitment to data quality and reliability. The experimental analysis is also thorough and multifaceted, covering diverse dimensions such as model type, user instruction form, and UI library, while providing rich error categorization and visual examples—including runtime failures, logical inconsistencies, and UI load analyses—that give readers an intuitive understanding of model behavior.

**Weaknesses:**

The overall task scale is relatively small—the real-world benchmark includes only 50 tasks (35 for Excel and 15 for Word). Although the authors emphasize their complexity, this limited number constrains the generalizability of conclusions about foundation models’ capabilities in complex API and UI generation.

The evaluation process also relies heavily on automated AI assessment with minimal human involvement. From functionality testing to UI scoring, most stages are conducted automatically. The functional tests use Playwright scripts generated by another LLM to simulate user interactions, which cannot fully capture usability issues that real users might face, such as unintuitive layouts or mismatched interaction logic. Similarly, the UI and UX assessments depend on a vision-based multimodal model to evaluate layout, readability, and cognitive load—criteria that may diverge significantly from those used by human UX experts. This raises concerns about the reliability and validity of the evaluation results.

The study focuses primarily on one-shot widget generation while overlooking interactive refinement. Given that even the strongest models achieve only around a 50% success rate, it seems unrealistic to expect perfect one-shot generation. A more practical evaluation would involve multi-turn collaboration between users and AI to iteratively debug and improve widgets. Although the paper briefly mentions “follow-up” settings, it does not fully explore this human–AI repair process, which arguably represents the most valuable and realistic aspect of dynamic widget generation.

**Questions:**

1. The reported 50% success rate, even for the strongest models such as GPT-4o and GPT-4.1 on complex Office applications, seems low from a practical standpoint. In real-world settings, a system that fails in one out of two attempts would be difficult to deploy. It would be useful to understand how the authors interpret this one-shot generation paradigm in relation to the actual usability and robustness of LLM-based agents.

2. The benchmark also omits the evaluation of interactive repair, despite acknowledging in its discussion that real users often engage in multi-turn dialogue with the model to iteratively refine or debug widgets. This lack of assessment for human–AI collaboration leaves a significant gap in understanding the models’ practical problem-solving abilities.

3. The description of the “follow-up” scenario is somewhat ambiguous. As currently presented, it seems to test whether a successfully generated widget can generalize to a second task, rather than whether a model can recover from an initial failure using user feedback such as “the button click caused an error.” A clearer definition of this setting would strengthen the paper’s claims about generalization and adaptability.

---

> ### Author Response · Authors · 2025-11-21
> **Author response (1/2)**
>
> Thank you for your feedback and comments. We hope the below response addresses your concerns and please let us know if you have any further questions.
>
> **Size of the dataset:**
>
> While our real-world applications dataset contains only 50 main tasks (each paired with two test scenarios), it represents a highly diverse, and carefully curated collection. These tasks span a broad range of Excel and Word APIs—covering objects such as tables, charts, pivot tables, shapes, comments, ranges, paragraphs, fonts, conditional format, formulas, content controls, and images —and exercise approximately **240 Excel APIs** (functions and properties) and **85 Word APIs**. Importantly, each task requires the generation of both high-quality init_code and robust verif_scripts, which is the primary bottleneck in scaling this dataset. Verification of real-world application states is non-trivial: Office objects often involve rich, nested structures and property-based conditions that go beyond simple string or array comparisons. Because of this, every task in our dataset needs extensive human testing, filtering, and cleanup to eliminate false positives and false negatives and ensure reliability. We filtered and verified the final set from 200 examples we discovered online.
>
> To complement this dataset and broaden coverage of data types (e.g., dates, nested lists, hierarchical structures), we introduce the synthetic dataset. While these synthetic tasks are easier in terms of functional correctness—especially for larger models—they are still very useful. They serve as "unit tests" for us to systematically study fine-tuning strategies for smaller models, and insights from these experiments can transfer to improving performance on real-world tasks. Synthetic tasks also expose LLM challenges with widget designs and user interactions, which again helps us iterate on and improve usability across both real-world and synthetic settings.
>
> **Evaluation process relies heavily on automated AI assessment with minimal human involvement:**
>
> The evaluation process is fully automatic by design. At the current performance level—around 50% success on the Excel/Word widget tasks--there is significant development needed for LLMs to reliably solve the widget generation task, which in turn requires an automated evaluation pipeline to support rapid development and ablation studies.
>
> **One-shot generation and low success rate for practical applications:**
>
> This is exactly the challenge we would like to highlight from our evaluation! From our observation, generating a plausible looking interface (only the UI) does not guarantee a functional widget that can be reliably used for end users. We agree that evaluation results show that there is still a gap for directly deploying the model for end users to expect one-shot success.
>
> We also note that most of our tasks are designed to be solvable in a single shot, provided the model can accurately reason about user interfaces and the surrounding application context from common sense, app/UI knowledge, and available APIs. We expect that as model capabilities improve, future versions will achieve much higher success rates under this one-shot setting.
>
> However, even at this performance level, we envision that LLMs can still be used effectively for developers or advanced users to create functional widgets in an interactive setting where they can iteratively refine widgets as they observe widget behaviors. The particular benefit of generated widget is that once they are created, they can be used for the same parameterized tasks in future without additional human/AI intervention, thus the initial efforts to interactively refine the widgets would not prevent adoption of the approach. At the same time, we also think our benchmark can be used as a metric to effectively monitor the progress of LLMs for widget generation tasks.

---

> > ### Author Response · Authors · 2025-11-21
> > **Author response (2/2)**
> >
> > **Interactive repair:**
> >
> >
> > Based on the reviewer's suggestion, we performed an additional experiment to see the impact of interactive repair. Using the execution results, verification script outputs, and screenshots, we used a multi-modal agent to explain what is wrong with the current widget. We then used this feedback to guide the original model to fix its errors.
> >
> > For this experiment, on the Excel dataset, we found that o4-mini model with tool calling **improves success rate from 54% to 74% with a single step repair**. For example, for a task that asks to "Modify the straight line in Excel by adding arrowheads to both ends", the initially generated widget has a dropdown for selecting the line shape (straight, curved, etc), but failed to populate the options correctly, making it impossible for the user (or simulated user) to select 'StraightLine'. Upon receiving this feedback, the model successfully fixed the issue in the next iteration. In other cases, the model was able to successfully recover from runtime errors with feedback. However, we also observed scenarios where the model fixed the identified issue but introduced a new one, highlighting that some tasks may require multiple rounds of repair.
> >
> >
> > **Definition of follow-up scenario:**
> >
> > Yes, the "follow-up" scenario in our case tests whether a widget can be used to perform multiple actions one-after the other (which sometimes requires remembering what was done before and undoing the changes performed in the previous action).

---

> > > ### Comment · Reviewer_1cY8 · 2025-11-25
> > >
> > > Thank you for your clarification. I will maintain my scores.

---

### Official Review · Reviewer_HSFF · 2025-10-30

**Soundness:** 2
**Presentation:** 3
**Contribution:** 2
**Rating:** 4
**Confidence:** 4

**Summary:**

This paper presents a benchmark named WidgetEval for code generation. Each task in WidgetEval is to generate a piece of code that can generate a widget in a specific application under a specific state to satisfy some natural language description. In particular, there are two sources for WidgetEval. The first source is tasks for Excel and Word, and the second source is tasks adapted from NLI (a semantic parsing dataset). This paper also presents empirical evidence for some LLMs to deal with WidgetEval.

**Strengths:**

1. To my knowledge, this is the first benchmark for generating widgets.
2. The empirical evidence demonstrates that the tasks for Excel and Word are challenging for existing LLMs.

**Weaknesses:**

1. It is unclear how the proposed benchmark is connected to UI development in real world. The proposed benchmark seems to be launching some interfaces under specific program states. Therefore, it is unclear whether the proposed benchmark is helpful for improving LLM-based techniques for UI development.
2. The tasks in WidgetEval are ultimately in the form of code generation. But it is unclear how WidgetEval differs from existing code generation benchmarks for evaluating LLMs. That is to say, it is unclear whether an LLM that is good at generating code would also be good at generating widgets for WidgetEval.
3. The synthetic part may be less useful. Since the synthetic tasks are adapted from a semantic parsing dataset, it is unclear how these tasks can provide additional information besides the original semantic parsing tasks when used for evaluating LLMs.

**Questions:**

1. How can the proposed benchmark benefit UI development that existing approaches to GenUI primarily target.
2. Do you have evidence that WidgetEval can provide additional information that typical code generation benchmarks when evaluating LLM-based approaches to GenUI.
3. Do you have evidence that LLMs would perform differently for the semantic parsing tasks and the widget generation tasks adapted from these semantic parsing tasks?

---

> ### Author Response · Authors · 2025-11-21
> **Author response**
>
> Thank you for your feedback and comments. We hope the below response addresses your concerns and please let us know if you have any further questions.
>
> **How WidgetEval helps UI development?:**
>
> Real world UI development (app builders / UX agents that are developed for non-coders to create frontend applications such as v0, Lovable, Replit) is not just about designing static layouts but also to design responsive interfaces that function reliably based on the content and structure of the host application. These context-dependent interfaces are the main motivation for our widgets dataset. Our widget tasks are evaluating the model’s ability to design and produce functional, state-aware UIs that solve individual user problems. A full end-to-end application typically requires orchestrating many such widgets in a composable manner. Therefore, in our opinion, widget generation ability is an essential component for LLM based end-to-end UI development, and our benchmark directly targets this capability and provides a principled way to measure and improve it.
>
> **WidgetEval vs code generation benchmarks:**
>
>
> Code generation is only the final step in the much more complex process needed for widget generation. Mainly, widget generation requires interpreting user intent in the context of the current application, inferring appropriate user interactions, designing a functional UI, and finally generating the code that puts all the above together. In fact, most code-generation datasets provide full, explicit specifications of the desired program behavior. In contrast, widget tasks do not include complete specifications. The model must infer missing details using common sense, knowledge of the application context, and understanding of the underlying APIs.
>
> Additionally, unlike typical coding tasks that are evaluated on program outputs, widget evaluation requires user interaction with the UI to surface widget errors and usability issues. Our approach provides test cases and simulates human interactions to understand interaction issues and widget functionality. This enables automatic evaluation of LLM's ability to generate functional UI besides basic coding evaluations.
>
> In fact, as we show in table one, models with higher code generation ability are not guaranteed to generate widgets that are more reliable and usable. Among close-sourced models, the performance on the widget tasks is different from code tasks. For example, o4-mini is better than GPT-4o on widget tasks (Table 1) whereas GPT-4o is better on HumanEval, a commonly used code dataset (https://openai.com/index/gpt-4o-mini-advancing-cost-efficient-intelligence/).   This indicates functional widget generation has a different distribution than general coding that is worth studying with its own benchmarks
>
> **WidgetEval vs semantic parsing tasks:**
>
> Semantic parsing tasks are substantially easier: they require identifying the correct APIs to make a single concrete change to the application state. In contrast, widget tasks require more complex reasoning—designing generalizable logic that supports multiple user inputs, exposing appropriate UI options, and coordinating UI behavior with underlying application objects.
>
> In fact, on the semantic parsing tasks in the Calendar domain, LLaMA-3.1-70B achieves an almost perfect success rate of 95%. However, the same model achieves only 59% on the widget-generation tasks (Table 3).

---

> > ### Comment · Reviewer_HSFF · 2025-11-25
> > **Response to the authors' rebuttal**
> >
> > After reading your rebuttal, my main concern remains. In your rebuttal, you seem to say that you aim at the code generation part in UI development. If so, the proposed dataset is still a dataset for evaluating code generation, but under a special circumstance. Thus, I think one of the primary tasks is to convince readers that new dataset brings vast additional values to existing code generation datasets. But I am not fully convinced by the uniqueness and importance of the proposed datasets.

---

### Official Review · Reviewer_EHCv · 2025-10-31

**Soundness:** 3
**Presentation:** 3
**Contribution:** 3
**Rating:** 6
**Confidence:** 3

**Summary:**

The paper investigates the generation of application widgets on-the-fly. Widgets are a particular type of code generation that usually involves interacting with a UI and wrapping APIs that are made available by the applications. This type of generation is challenging because it requires understanding the application context + APIs, and reasoning about the UI/UX. The paper puts together a benchmark of widget generation tasks that focus on applications such as Excel and Word, and a series of tasks that focus on synthetic applications. While the tasks are not that hard, the performance of the models, even the closed-source ones, are generally lower than 50%.

**Strengths:**

Putting together a relevant dataset that could be used by the research community to work at the intersection of UI manipulation and code generation

**Weaknesses:**

The paper is in a niche domain and it is not clear how many would use/benefit from the benchmarks introduced in this work.

**Questions:**

Can you elaborate on the utility of the widgets? How would they be used? How do they simplify user experience?

---

> ### Author Response · Authors · 2025-11-21
> **Author response**
>
> Thank you for your feedback and comments. We hope the below response addresses your concerns and please let us know if you have any further questions.
>
> **Niche domain:**
>
> We agree that there is limited prior work in this domain. However, we consider this to be a strength of the work and a contribution to its novelty. Creating benchmarks and evaluation systems for complex and nuanced applications like stateful widgets is challenging and contributes to the lack of work in this area.  We hope this work can help support and grow this area, which can have a significant impact on how users interact with computers.
>
> Our work also relates to app builder / UX agents that are developed for non-coders to create frontend applications (e.g., v0, Lovable, Replit). And as far as we know, there have not been automatic evaluation protocols for full functionality of widgets being created (prior evaluations like LLM arena are only based on UI quality, not widget functionality), and our benchmark fills this gap.
>
> **Utility of dynamic widgets:**
>
>
> The use of dynamic widgets is supported by multiple lines of work featuring user evaluation from the HCI research community, including:
>
> DynaVis: Dynamically Synthesized UI Widgets for Visualization Editing | Proceedings of the 2024 CHI Conference on Human Factors in Computing Systems
>
> Dynamic Prompt Middleware: Contextual Prompt Refinement Controls for Comprehension Tasks | Proceedings of the 4th Annual Symposium on Human-Computer Interaction for Work
>
> https://ieeexplore.ieee.org/document/10714542/
>
> “What It Wants Me To Say”: Bridging the Abstraction Gap Between End-User Programmers and Code-Generating Large Language Models | Proceedings of the 2023 CHI Conference on Human Factors in Computing Systems
>
> Dynamic widgets allow users to create reusable widgets in applications (like Excel, Word) to help them customize their experiences in the tool for repetitive tasks. For example, if the user frequently needs to change colors of all paragraph headers their document, they can create a dynamic widget with AI that can automatically control colors of paragraph headers. This way, they don't have to query LLMs every time they want to change the style (which requires additional conversation and clarification overhead and may lead to inconsistencies) or manually change paragraph color one by one with default built-in widgets. The dynamic widgets allow users to customize their interface within the "create once reuse over time" experience.
>
> Dynamic widgets also help bridge the 'abstraction gap' where users do not know the correct language to invoke the behavior they want but can explore or modify a generated widget to perform the action. It is useful when users are unaware of the capabilities of a system and cannot discover the right widget combinations to solve their tasks in hand. Dynamic widgets can also help users learn the capabilities of a system as they create customized widgets based on their tasks.

---

> > ### Comment · Reviewer_EHCv · 2025-11-21
> >
> > Thank you for your clarification. I will maintain my scores.

---

### Official Review · Reviewer_ELhL · 2025-11-02

**Soundness:** 3
**Presentation:** 3
**Contribution:** 3
**Rating:** 4
**Confidence:** 4

**Summary:**

The paper addresses the task of automatic widget creation for simple, repetitive tasks in existing UI applications (e.g. word and excel). The task is well motivated. Building these widgets can be tedious for users but the widgets themselves can be quite useful, since they provide intuitive interfaces that allow users to do many repetitive tasks.

The paper introduces a benchmark created out of real and synthetic tasks and presents an evaluation some strong closed and open source models. The analysis sheds some light into the failure modes of these models on the widget generation task and highlight the difficulty of the problem.

**Strengths:**

The main strengths of the paper are the following:

**Task Importance and Impact**  The task is well-motivated.

- Widgets can greatly increase productivity and improve user experience. The choice of Word and Excel, which are some of the most widely used applications, mean that progress on this task can have broad impact.

- The task is challenging along many dimensions. It not only requires code generation, and state tracking but also aspects of UI design, and ability to generalize to future needs.

**Benchmark Construction** The major design elements in the construction of the benchmark are sound.
- It covers the main types of use cases for benchmark creation (specific task, generalizing to all tasks of the same type, and more generic description of widget capabilities).
- While the instances are sourced using LLMs, adequate controls (including manual inspection) are employed to ensure that the tasks are well-defined.
- It also includes a synthetic dataset, which is automatically generated

**Evaluation** The central pain point in tasks such as these is evaluation. We need to check if the widget is functionally correct and not just compare it to reference code.

- The benchmark provides such a state-based evaluation to assess the effectiveness the automatically generated widgets. Playwright based evaluation allows for simulating user interaction with the widget. The generated widget is thus executed using the simulated user and the final state after widget execution is compared against an expected state.

- The evaluation is reproducible -- it provides resettable states via init scripts that ensure the state of the environment (i.e. document or spreadsheet) is the same for every run.

- The evaluation also covers UI aspects in addition to functional correctness. This evaluation considers (VLM judge) scores on five different dimensions covering ease of use, and layout aspects.

**Weaknesses:**

There are two key weaknesses in the paper.

**Size of the dataset** There are only 50 main instances in total from the existing applications (e.g. Excel and Word applications). This is too small a dataset for a community wide benchmark. The synthetic dataset while useful as a supplement appears to be substantially easier (and different) from the main dataset. The details of this dataset construction are sparse. In particular, it is difficult to assess whether this is a high quality dataset since there isnt much description of what kinds of quality control was done on this dataset and no indication or discussion as to whether the tasks would benefit from widgets in the same way as the original tasks.

**Analysis**
- The results are presented mostly as just observations of what was found.
- The error categorization is useful but is at a higher level. The analysis doesnt provide insights into why the models failed in these cases.
-

**Questions:**

- Can you please provide some justification on why the synthetic dataset represents a meaningful collection of tasks that can benefit from widget generation? Also please provide some more details on what kind of quality control was done on these.
- I am not quite sure about the number of instances in the training datasets. Are there only 50 instances in the main dataset?

---

> ### Author Response · Authors · 2025-11-21
> **Author response**
>
> Thank you for your feedback and comments. We hope the below response addresses your concerns and please let us know if you have any further questions.
>
> **Size of the dataset**:
>
> While our real-world applications dataset contains only 50 main tasks (each paired with two test scenarios), it represents a highly diverse, and carefully curated collection. These tasks span a broad range of Excel and Word APIs—covering objects such as tables, charts, pivot tables, shapes, comments, ranges, paragraphs, fonts, conditional format, formulas, content controls, and images —and exercise approximately **240 Excel APIs** (functions and properties) and **85 Word APIs**. Importantly, each task requires the generation of both high-quality init_code and robust verif_scripts, which is the primary bottleneck in scaling this dataset. Verification of real-world application states is non-trivial: Office objects often involve rich, nested structures and property-based conditions that go beyond simple string or array comparisons. Because of this, every task in our dataset needs extensive human testing, filtering, and cleanup to eliminate false positives and false negatives and ensure reliability. We filtered and verified the final set from 200 examples we discovered online.
>
> To complement this dataset and broaden coverage of data types (e.g., dates, nested lists, hierarchical structures), we introduce the synthetic dataset. While these synthetic tasks are easier in terms of functional correctness—especially for larger models—they are still very useful. They serve as "unit tests" for us to systematically study fine-tuning strategies for smaller models, and insights from these experiments can transfer to improving performance on real-world tasks. Synthetic tasks also expose LLM challenges with widget designs and user interactions, which again helps us iterate on and improve usability across both real-world and synthetic settings.
>
> Overall, the real-world dataset is intentionally small but deeply vetted and representative of real application complexity, while the synthetic tasks play a complementary role in scaling coverage, allowing the community to experiment with finetuning strategies, and improving LLM-based widget design.
>
>
> **Synthetic dataset and its meaningfulness:**
>
>
> The synthetic domain originates from the work by Givoli and Reichart (Zero-shot semantic parsing for instructions).
>
> "We collected the dataset by presenting human annotators with visualizations of initial and desired state pairs. The annotators were then asked to write an English instruction that can be executed in order to transfer the application from the initial state to the desired state."
>
> givoli/TechnionNLI: This repository contains the dataset and code for the ACL 2019 paper "Zero-Shot Semantic Parsing for Instructions" by Ofer Givoli and Roi Reichart.
>
> For an example image see: https://github.com/givoli/TechnionNLI/raw/master/docs/images/task.jpg?raw=true
>
> As part of this work, we performed additional quality control on the selected domains. Using a powerful model (gpt4) we generated code solutions. For cases where GPT-4o was unable to generate a correct solution after multiple trials, we inspected the datapoint. In a few cases the utterance was misleading, and we pruned that from the set. As a result, the remaining points were either solvable by GPT-4o or inspected by us to ensure the example made sense.
>
> We believe that the selected domains (calendar, messenger, file system) are well suited to widget tasks. Each domain is commonly associated with GUI applications (email clients, chat clients, file browsers), and therefore we think dynamic generation of contextual GUIs (widgets) is appropriate for these domains and tasks. Furthermore, many of these tasks involve destructive actions (moving / deleting a file, calendar entry) and having a visual confirmation of the proposed action can provide additional interpretability. Lastly, as we describe in our response to reviewer EHCv, there are general HCI principles that underpin the benefit of widget generation, which extends to this task.

---

> > ### Comment · Reviewer_ELhL · 2025-11-21
> > **Appreciate the response but my concerns remain.**
> >
> > Thanks for further clarifying your dataset construction process and the nature of the resulting dataset. While I understand the complexity and the difficulty of the dataset you have created, the size of the dataset is still a huge concern. The overall number of tasks is 50. It is really small as a test set to draw any meaningful statistical comparisons once multiple teams start working on this dataset.
> >
> > I understand that the synthetic dataset is constructed from human inputs but it is not similar in complexity to the primary dataset and has substantially higher model performance (as shown in your results).
> >
> > While I understand the quality of the datasets a little better my concerns remain.

---

### Meta-Review · Area_Chair_3Ho4 · 2026-01-07

**Summary:**

This paper introduces WidgetEval, a benchmark for evaluating LLMs on dynamic widget generation within existing applications, primarily Microsoft Excel and Word, complemented by a synthetic task suite adapted from prior semantic parsing datasets. The work aims to assess whether models can translate natural language instructions and application context into functional, state-aware UI widgets, and proposes an automated, execution-based evaluation framework.

All reviewers agree that the problem setting is well motivated and practically relevant. Automatically generating reusable widgets for repetitive tasks in widely used applications could significantly improve user productivity and represents a meaningful intersection of code generation, UI/UX reasoning, and application context understanding. However, there are also several major concerns raised:

1. Limited scale of the real-world dataset. The benchmark contains only 50 core Office tasks, which multiple reviewers argue is too small to serve as a community-wide evaluation standard or to support robust statistical comparisons as more methods emerge. While the authors provide a detailed justification—highlighting the diversity of APIs covered, the complexity of verification, and the significant human effort required—reviewers largely remain unconvinced that this mitigates the risks associated with such a small test set.

2. The synthetic dataset is generally viewed as a weaker component. Reviewers agree it is substantially easier than the real-world tasks and question how well it captures the same complexity or practical value of widget generation. Although the authors argue that these tasks act as unit tests and support fine-tuning and analysis, several reviewers feel the gap in difficulty and realism limits their usefulness as a proxy for real applications.

3. Another point of contention is the positioning of the benchmark relative to existing code generation and GenUI work. Some reviewers remain unconvinced that WidgetEval provides sufficiently distinct signals beyond specialized code generation under constraints, despite the authors’ arguments about incomplete specifications, UI reasoning, and interaction-based evaluation. Relatedly, concerns are raised about the reliance on automated evaluation (LLM-generated test scripts and VLM-based UI judgments) and the limited involvement of human UX assessment.

4. Reviewers note that the focus on one-shot generation—with success rates around 50%—limits immediate practical applicability. While the authors’ additional experiments on interactive repair are appreciated and show promising gains, reviewers felt this aspect remains underexplored relative to its real-world importance.

Overall, this submission sits near the borderline. Reviewers’ scores span from marginally below to marginally above the acceptance threshold. Given that the concerns of two reviewers still remain after rebuttal, I believe strengthening the paper’s positioning, more explicitly acknowledging the limitations of scale, and clarifying the intended role of WidgetEval relative to existing code and UI benchmarks would help improve the paper for next round of submission.

**Reviewer Concerns:**

see above

**Reviewer Scores:**

see above

---

### Decision · Program_Chairs · 2026-01-26

Reject